# mAggretriever: A Simple Yet Effective Approach to Zero-Shot Multilingual Dense Retrieval

**Sheng-Chieh Lin**[1*], **Amin Ahmad**[2], **Jimmy Lin**[1]

[1] University of Waterloo    [2] Vectara

{s269lin,jimmylin}@uwaterloo.ca, amin@vectara.com

## Abstract

Multilingual information retrieval (MLIR) is a crucial yet challenging task due to the need for human annotations in multiple languages, making training data creation labor-intensive. In this paper, we introduce mAggretriever, which effectively leverages semantic and lexical features from pre-trained multilingual transformers (e.g., mBERT and XLM-R) for dense retrieval. To enhance training and inference efficiency, we employ approximate masked-language modeling prediction for computing lexical features, reducing 70–85% GPU memory requirement for mAggretriever fine-tuning. Empirical results demonstrate that mAggretriever, fine-tuned solely on English training data, surpasses existing state-of-the-art multilingual dense retrieval models that undergo further training on large-scale MLIR training data. Our code is available at https://github.com/castorini/dhr.

## 1 Introduction

Fine-tuning a pre-trained transformer has proven to be highly effective in many tasks of natural language processing, including information retrieval (IR). Despite its success, the recent state-of-the-art dense retrieval (DR) models (Ni et al., 2022; Lin et al., 2023a) predominantly focus on English. This bias arises from the fact that constructing an effective DR model requires a substantial amount of annotated training data, which is predominantly available in English datasets (Bajaj et al., 2016; Kwiatkowski et al., 2019). This makes it challenging for users of low-resource languages to benefit from the recent IR progress.

To address this issue, researchers have explored leveraging pre-trained multilingual transformers, such as mBERT (Devlin et al., 2019) and XLM-R (Conneau et al., 2020), which exhibit promising language transferability even when fine-tuned on

English datasets alone (Zhang et al., 2023b). However, Izacard et al. (2022) emphasize the importance of contrastive pre-training on multilingual corpora to achieve improved language transferability in IR. Other approaches utilize multilingual parallel corpora or translation pairs for contrastive pre-training (Feng et al., 2022) or fine-tuning (Reimers and Gurevych, 2020; Bonifacio et al., 2021). However, these solutions are viable only when significant computational resources or multilingual parallel data are available.

Recently, Lin et al. (2023b) demonstrate that the existing DR models solely using [CLS] (or averaged pooling) do not fully exploit the capability from pre-trained transformers for retrieval. Lin et al. (2023b) propose Aggretriever by combining the semantic and lexical features from the respective two components, [CLS] and masked language modeling (MLM) prediction, which shows superior effectiveness on diverse English retrieval tasks. This inspires us to ask the question: Can Aggretriever benefit multilingual retrieval?[1]

Extending Aggretriever to support multilingual retrieval poses challenges due to increased computation costs for extracting lexical features from the MLM component as the vocabulary size of the pre-trained model grows, making fine-tuning with limited resources challenging. In this work, we propose two simple approaches to approximate MLM prediction to extract lexical features from multilingual pre-trained transformers, making the training of mAggretriever possible in one GPU within 40 GBs of memory. Remarkably, mAggretriever exhibits strong retrieval capability across multiple languages despite being fine-tuned on English data.

The paper is structured as follows: we begin by providing background information on standard DR and Aggretriever. We then present our proposed approaches to tackle the computational challenges

---

*Work done during Sheng-Chieh's internship at Vectara.

[1]In this paper, we refer multilingual retrieval to monolingual retrieval across multiple languages.

involved in extending Aggretriever to support multilingual retrieval. Subsequently, we compare mAggretriever with other state-of-the-art multilingual DR models and explore the potential for extending mAggretriever to enable cross-lingual retrieval.

## 2 Background

**Dense Retrieval.** Given a query with sequential tokens $q = (\texttt{[CLS]}, q_1, \cdots q_n)$, our task is to retrieve a list of passages to maximize some ranking metric such as nDCG or MRR. Standard dense retrieval (DR) models (Reimers and Gurevych, 2019; Karpukhin et al., 2020) based on pre-trained language models encode queries and passages as low dimensional [CLS] vectors with a bi-encoder architecture and use the dot product between the encoded vectors as the similarity score:

$$\text{sim}_{\text{CLS}}(q, p) \triangleq \mathbf{e}_{q_{\texttt{[CLS]}}} \cdot \mathbf{e}_{p_{\texttt{[CLS]}}}, \quad (1)$$

where $\mathbf{e}_{q_{\texttt{[CLS]}}}$ and respective query and passage $\mathbf{e}_{p_{\texttt{[CLS]}}}$ are the [CLS] vectors at the last layer of a pre-trained language model (e.g., BERT).

**Aggretriever.** In addition to using [CLS] vectors to capture semantic textual features, Lin et al. (2023b) further propose to capture lexical textual features from the pre-trained MLM prediction head by projecting each contextualized token embedding $\mathbf{e}_{q_i}$ into a high-dimensional vector in the wordpiece lexical space:

$$\mathbf{p}_{q_i} = \text{softmax}(\mathbf{e}_{q_i} \cdot \mathbf{W}_{\text{mlm}} + \mathbf{b}_{\text{mlm}}), \quad (2)$$

where $\mathbf{e}_{q_i} \in \mathbb{R}^d$, $\mathbf{W}_{\text{mlm}} \in \mathbb{R}^{d \times |V_{\text{wp}}|}$, and $\mathbf{b}_{\text{mlm}} \in \mathbb{R}^{|V_{\text{wp}}|}$ are the weights of the pre-trained MLM linear projector, and $\mathbf{p}_{q_i} \in \mathbb{R}^{|V_{\text{wp}}|}$ is the $i$-th contextualized token represented by a probability distribution over the BERT wordpiece vocabulary, $V_{\text{wp}}$. Weighted max pooling is then performed over the sequential representations $(\mathbf{p}_{q_1}, \mathbf{p}_{q_2}, \cdots, \mathbf{p}_{q_l})$ to obtain a single-vector lexical representation:

$$\mathbf{v}_q[v] = \max_{i \in (1, 2, \cdots, l)} w_i \cdot \mathbf{p}_{q_i}[v], \quad (3)$$

where $w_i = |\mathbf{e}_{q_i} \cdot \mathbf{W} + \mathbf{b}| \in \mathbb{R}^1$ is a positive scalar and $v \in \{1, 2, \cdots, |V_{\text{wp}}|\}$; $\mathbf{W} \in \mathbb{R}^{d \times 1}$ and $\mathbf{b} \in \mathbb{R}^1$ are trainable weights. Note that the scalar $w_i$ for each token $q_i$ is essential to capture term importance, which $\mathbf{p}_{q_i}$ alone cannot capture since it is normalized by softmax. Note that the [CLS] token embedding is excluded since it is used for

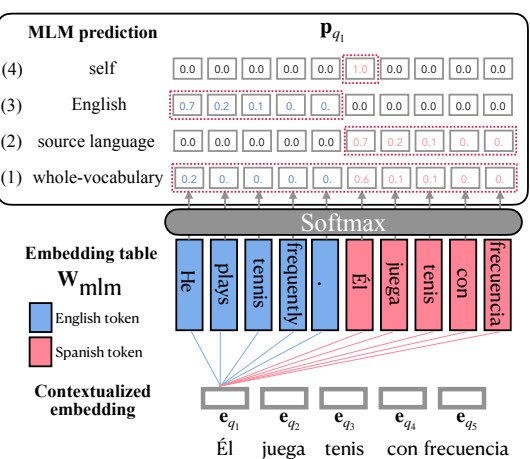

Figure 1: Illustration of MLM prediction for the input token $q_1$. (1) Whole-vocabulary prediction considers all languages while target-language prediction only considers the tokens in one language; e.g., (2) source language (Spanish) or (3) English. (4) Self prediction assigns probability of one to itself. 0. denotes less than 0.1.

next-sentence prediction during pre-training; thus, does not carry much lexical information.

The wordpiece lexical representation $\mathbf{v}_q$ is then compressed, without supervision, into low-dimensional vector $\mathbf{agg}_q^\star$ and concatenated with the [CLS] vector. The similarity score between a given $q$–$p$ pair is computed with their dot product:

$$\text{sim}(q, p) \triangleq (\mathbf{e}_{q_{\texttt{[CLS]}}} \oplus \mathbf{v}_q) \cdot (\mathbf{e}_{p_{\texttt{[CLS]}}} \oplus \mathbf{v}_p) \quad (4)$$
$$\approx (\epsilon_{q_{\texttt{[CLS]}}} \oplus \mathbf{agg}_q^\star) \cdot (\epsilon_{p_{\texttt{[CLS]}}} \oplus \mathbf{agg}_p^\star),$$

where $\mathbf{e}_{q_{\texttt{[CLS]}}}$ and $\mathbf{v}_q$ capture semantic and lexical textual features from BERT, respectively. Following Lin et al. (2023b), we linearly project $\mathbf{e}_{q_{\texttt{[CLS]}}}$ into 128 dimension and compress $\mathbf{v}_q$ into 640 dimension; i.e., $\epsilon_{q_{\texttt{[CLS]}}} \in \mathbb{R}^{128}$ and $\mathbf{agg}_q^\star \in \mathbb{R}^{640}$. We refer readers to Lin et al. (2023b) for the details of compressing $\mathbf{v}_q$ into $\mathbf{agg}_q^\star$.

## 3 Our Approach

In this work, we extend Aggretriever to multilingual retrieval, mAggretriever. However, directly applying Eq. (2) to multilingual pre-trained MLM heads over the whole vocabulary increases computation cost for both training and inference due to large vocabulary size $|V_{\text{wp}}|$ in MLM projector, $\mathbf{W}_{\text{mlm}}$ (Nair et al., 2022; Lassance, 2023). For example, the mBERT and XLM-R have respective vocabulary size of 120K and 250K (vs BERT's 35K) in the MLM projector. To address the issue, we propose two computationally efficient approaches to approximate MLM prediction in Eq. (2).

Table 1: Training and inference cost comparisons on MS MARCO with XLM-R base backbone.

| Models | vector components | | MLM prediction | training cost | | query encoding latency | |
|---|---|---|---|---|---|---|---|
| | `[CLS]` | **agg**$^\star$ | | GPU memory | total time | CPU | GPU |
| (1) XLM-R$_{\text{CLS}}$ | 768 dim. | 0 dim. | - | 21 GBs | 5.5 hrs | 105 ms/q | 9.1 ms/q |
| (2) XLM-R$_{\text{AGG}}$ | | | whole vocabulary | OOM (140 GBs) | - | 400 ms/q | 15.1 ms/q |
| (3) XLM-R$_{\text{AGG}}^{\text{tg}}$ | 128 dim. | 640 dim. | target language | 39 GBs | 9.5 hrs | 248 ms/q | 11.4 ms/q |
| (4) XLM-R$_{\text{AGG}}^{\text{self}}$ | | | self | 21 GBs | 5.5 hrs | 123 ms/q | 9.1 ms/q |

**Target-language prediction.** One intuitive approach is to compute the probability distribution over the target-language token set of interest, denoted as $V_{\text{wp}}^{\text{tg}}$, by replacing $\mathbf{W}_{\text{mlm}}$ and $\mathbf{b}_{\text{mlm}}$ in Eq. (2) with $\mathbf{W}_{\text{mlm}}^{\text{tg}}$ and $\mathbf{b}_{\text{mlm}}^{\text{tg}}$, respectively.

$$\begin{cases} \mathbf{W}_{\text{mlm}}^{\text{tg}}[:,v] = \mathbf{W}_{\text{mlm}}[:,v]; \\ \mathbf{b}_{\text{mlm}}^{\text{tg}}[v] = \mathbf{b}_{\text{mlm}}[v], & \text{if } v \in V_{\text{wp}}^{\text{tg}} \\ \mathbf{W}_{\text{mlm}}^{\text{tg}}[:,v] = \overrightarrow{0}; \mathbf{b}_{\text{mlm}}^{\text{tg}}[v] = 0, & \text{otherwise.} \end{cases}$$

From the above equation, we only have to compute the matrix multiplication and softmax among $|V_{\text{wp}}^{\text{tg}}|$ instead of $|V_{\text{wp}}|$ tokens. This approach assumes that only the tokens corresponding to the target language have responses when conducting MLM prediction; i.e., $\mathbf{p}_{q_i}[v] = 0$ if $v \notin V_{\text{wp}}^{\text{tg}}$.

**Self prediction.** Since BERT MLM pre-training task includes recovering the unmasked token itself, to further save computation cost, we may approximate MLM prediction by assigning a value of one to the token $q_i$ itself, and zero to the others:

$$\mathbf{p}_{q_i} = x_j \in \{0,1\}^{|V_{\text{wp}}|} \text{ for } j \in \{\text{tok\_id}(q_i)\}. \quad (5)$$

The operation removes the computationally expensive matrix multiplication and softmax operation in Eq. (2). Note that, combining Eq. (3) and (5), the lexical representations, $\mathbf{v}_q$, from self prediction can be considered bag-of-word vectors with learned term weights.

Figure 1 illustrates whole-vocabulary, our proposed target-language and self MLM predictions. Table 1 compares the training and inference cost of the standard DR (row 1) and mAggretriever with different MLM prediction strategies on MS MARCO dataset using the backbone of XLM-R base. The training and query encoding settings are detailed in Section 4.2. We observe training with target-language (English in our case) MLM prediction reduces the GPU memory requirement compared to whole vocabulary (row 3 vs 2) while self prediction yields training and inference efficiency on par with standard DR (row 4 vs 1).

## 4 Experimental Setups

### 4.1 Datasets and Metrics

We use the MS MARCO passage ranking dataset introduced by Bajaj et al. (2016), comprising a corpus with 8.8M passages and 500K training queries. Model supervised language (English) retrieval effectiveness is evaluated on the 6980 (MARCO dev) queries from the development set with one relevant passage per query on average. Following the established procedure, we report MRR@10 and R@1000 as the metrics.

We evaluate model zero-shot retrieval effectiveness in other languages using MIRACL dataset introduced by Zhang et al., comprising around 77k queries over Wikipedia in 18 languages with over 700k high-quality relevance judgments by native speakers. We use the publicly available development queries and their relevance judgements in 15 languages (two surprise languages and English are excluded).[2] Following Zhang et al., we report macro averaged nDCG10 and R@100 over the 15 languages and list the full numbers in Appendix A.1.

Finally, we study how to conduct cross-lingual retrieval using mAggretriever on XQuAD-R dataset introduced by Roy et al. (2020), consisting of parallel queries and corpora with 11 languages.[2] We conduct retrieval using the queries with $X_Q$ language against the corpus with $X_C$ language and report the macro-averaged MAP@100 over all the cross-lingual combinations of the 110 language pairs ($X_Q \neq X_C$), and the other 11 monolingual combinations ($X_Q = X_C$).

Table 2 reports the data statistics of MIRACL and XQuAD-R. Note that the candidates of MIRACL are passages while the candidates of XQuAD-R are chunked sentences from XQuAD corpora. Note that although XQuAD corpora is the manually rewritten multilingual parallel corpora (Artetxe et al., 2020), the numbers of chunked sentences are

---

[2] Datasets: MIRACL, XQuAD-R

Table 2: MIRACL and XQuAD-R data statistics.

| language | ISO | MIRACL Dev | | XQuAD-R | |
|---|---|---|---|---|---|
| | | # queries | # candidates | # queries | # candidates |
| Arabic | ar | 2,869 | 2,061,414 | 1,190 | 1,222 |
| Bengali | bn | 411 | 297,265 | - | - |
| German | de | - | - | 1,190 | 1,276 |
| Greek | el | - | - | 1,190 | 1,234 |
| English | en | 648 | 32,893,221 | 1,190 | 1,180 |
| Spanish | es | 799 | 10,373,953 | 1,190 | 1,215 |
| Persian | fa | 632 | 2,207,172 | - | - |
| Finnish | fi | 1,271 | 1,883,509 | - | - |
| French | fr | 343 | 14,636,953 | - | - |
| Hindi | hi | 350 | 506,264 | 1,190 | 1,244 |
| Indonesian | id | 960 | 1,446,315 | - | - |
| Japanese | ja | 860 | 6,953,614 | - | - |
| Korean | ko | 213 | 1,486,752 | - | - |
| Russian | ru | 1,252 | 9,543,918 | 1,190 | 1,219 |
| Swahili | sw | 482 | 131,924 | - | - |
| Telugu | te | 828 | 518,079 | - | - |
| Thai | th | 733 | 542,166 | 1,190 | 852 |
| Turkish | tr | - | - | 1,190 | 1,167 |
| Vietnamese | vi | - | - | 1,190 | 1,209 |
| Chinese | zh | 393 | 4,934,368 | 1,190 | 1,196 |

different between languages. Also note that for each query, MIRACL has multiple relevant candidates while XQuAD only has one.

## 4.2 Implementation Details

**Models.** We apply mAggretriever to two 12-layer pre-trained multilingual models: (1) mBERT; (2) XLM-R.[3] We compare models fine-tuned solely using `[CLS]` vector and based on mAggretriever using whole-vocabulary MLM prediction with the subscripts "CLS" and "AGG", respectively, e.g., $mBERT_{CLS}$ and $mBERT_{AGG}$. We report the two variants of mAggretriever with target-language and self prediction; e.g., $mBERT_{AGG}^{tg}$ and $mBERT_{AGG}^{self}$. For target-language prediction, we train mAggretriever using English token prediction and run inference using the corresponding language of each corpus, and for whole-vocabulary prediction, we fine-tune with half of the batch size. In addition, we report the numbers of BM25 and mDPR from Zhang et al. as reference points, and the two state-of-the-art multilingual retrievers: (1) mContriever (Izacard et al., 2022), pre-trained on multilingual corpora with 29 languages and further fine-tuned on MS MARCO dataset;[3] (2) Cohere (API), whose numbers are copied from Kamalloo et al. (2023).[4]

**Training and Inference.** We train our models on a single A100 GPU with 80 GB memory for 6 epochs (around 100k steps) with learning rate

---

[3] Model checkpoints: mBERT, XLM-R, mContriever
[4] Cohere multilingual retrieval model

7e-6. Each batch includes 24 queries, and for each query, we randomly sample one positive and seven negative passages. All the negatives are sampled from the MS MARCO "small" triples training set, which is created using BM25. During training, we minimize the negative log likelihood as the standard contrastive loss and following Karpukhin et al. (2020), for each query, we consider all the (positive and negative) passages from the other triplets in the batch as in-batch negative samples. We set the maximum input length for the query and the passage to 32 and 128, respectively, at both training and inference stages for MS MARCO. For MIRACL and XQUAD-R, we use the maximum input length of 128 and 256 for the query and passage, respectively. Note that we lowercase all the queries and passages for mAggretriever.[5] We measure query encoding latency on the 6980 MS MARCO development queries with the batch size of 1 and single thread on a Linux machine with 12 Intel(R) Xeon(R) Gold 5317 CPU @ 3.00GHz and 88G of RAM.

**Target Token Set Construction.** For each language of corpus in MIRACL and XQuAD-R, we tokenize and lowercase all the passages and collect the unique tokens in the corpus as the target token set. For example, when fine-tuning on MS MARCO dataset, we use the token set built from MS MARCO corpus. While conducting target-language MLM prediction on MIRACL Arabic queries and corpus, we use the token set collected from Arabic corpus as our target token set. Note that self MLM prediction does not require collecting the token set for the target language.

## 5 Results

### 5.1 Results on MIRACL

Table 3 reports models' retrieval effectiveness on MS MARCO and MIRACL development queries. We first observe that mAggretriever, incorporating lexical features, not only outperforms its CLS counterpart in supervised English retrieval (MARCO), but also exhibits superior transferability to other languages (MIRACL) regardless of backbone. In addition, $mBERT_{AGG}$, without introducing contrastive pre-training on multilingual corpora, outperforms mContriever in MIRACL in terms of nDCG@10. Note that $mBERT_{AGG}$ and mContriever are both initialized from mBERT. We

---

[5] Our preliminary experiments on MS MARCO show that lowercase improves mAggretriever while degrades its CLS counterpart.

Table 3: Supervised (MARCO) English and zero-shot (MIRACL) multilingual retrieval effectiveness comparisons. Full numbers are listed in Appendix A.1.

| Models | MARCO Dev | | MIRACL Dev | |
|---|---|---|---|---|
| | English | | 15 lang. macro avg. | |
| | MRR@10 | R@1K | nDCG@10 | R@100 |
| (a) BM25 | 18.8 | 85.8 | 39.6 | 78.5 |
| (b) mDPR | 29.6 | 94.6 | 41.7 | 78.9 |
| (c) mContriever | 27.4 | **97.1** | 43.8 | 85.9 |
| (d) Cohere (API) | - | - | 50.1 | - |
| (1) mBERT$_{CLS}$ | 29.1 | 93.6 | 36.1 | 71.6 |
| (2) mBERT$_{AGG}$ | 34.3 | 95.8 | 44.4 | 79.5 |
| (3) mBERT$_{AGG}^{tg}$ | 34.5 | 96.1 | 44.4 | 80.1 |
| (4) mBERT$_{AGG}^{self}$ | 34.2 | 95.6 | 46.9 | 82.3 |
| (5) XLM-R$_{CLS}$ | 31.1 | 93.8 | 39.3 | 73.9 |
| (6) XLM-R$_{AGG}$ | 34.7 | 96.1 | 52.9 | **86.4** |
| (7) XLM-R$_{AGG}^{tg}$ | **35.0** | 96.2 | **53.3** | 86.0 |
| (8) XLM-R$_{AGG}^{self}$ | 35.0 | 96.0 | 53.3 | 86.3 |

Table 4: Zero-shot retrieval effectiveness on XQuAD-R. $X_Q$ ($X_C$) denotes the language of queries (corpus).

| Models | MLM target lang. | | XQuAD-R | |
|---|---|---|---|---|
| | query | corpus | $X_Q = X_C$ | $X_Q \neq X_C$ |
| | | | MAP@100 | |
| (1) XLM-R$_{CLS}$ | - | - | 73.1 | **57.5** |
| (2) XLM-R$_{AGG}$ | - | - | 77.4 | 41.8 |
| (3) | $X_Q$ | $X_C$ | **77.4** | 36.0 |
| (4) XLM-R$_{AGG}^{tg}$ | $X_C$ | $X_C$ | **77.4** | 44.9 |
| (5) | English | English | 73.5 | 51.7 |
| (6) XLM-R$_{AGG}^{self}$ | - | - | 77.3 | 36.2 |

hypothesize that mContriever's high recall (i.e., R@1K) comes from its pre-training on multilingual corpora with 29 languages.

Switching to XLM-R backbone, mAggretriever even outperforms Cohere (API). Furthermore, we notice that mAggretriever with XLM-R backbone improves over mBERT more than its CLS counterpart does. For example, in the case of MIRACL, XLM-R$_{AGG}^{tg}$ exhibits a significant improvement over mBERT$_{AGG}^{tg}$ from an nDCG@10 score of 44.4 to 53.3, whereas XLM-R$_{CLS}$ only sees a modest improvement over mBERT$_{CLS}$ from 36.1 to 39.3. This notable enhancement highlights mAggretriever's ability to effectively utilize a superior pre-trained language model.

Finally, compared to whole-vocabulary and proposed approximate MLM predictions, we observe that self prediction shows comparable and even strong language transferability. We hypothesize that MLM prediction learned from English data cannot transfer well to other languages. It is worth mentioning that compared to whole-vocabulary MLM prediction, the proposed approximate MLM prediction, target-language and self prediction, are advantageous for real-world deployment since they show almost no effectiveness drop (sometimes even better) but require far less training and inference cost as shown in Table 1.

## 5.2 Results on XQuAD-R

In Table 4, we directly apply XLM-R based models fine-tuned on MS MARCO to XQuAD-R dataset. In the experiment, we try different MLM prediction settings for XLM-R$_{AGG}^{tg}$. For example, instead of using respective query and corpus source language as the target language (row 3), we use the language corresponding to each corpus (row 4) or English (row 5) as target language for both queries and corpus. Note that rows 3 and 4 are the same when both queries and corpus are in the same language ($X_Q = X_C$).

We observe that mAggretriever shows relatively poor cross-lingual retrieval effectiveness ($X_Q \neq X_C$) compared to its CLS counterpart (rows 2,3,6 vs 1). When aligning the MLM prediction target language for queries and corpus, the cross-lingual retrieval effectiveness sees improvement (rows 3 vs 4,5). These results show that MLM prediction head potentially can be used as a translation layer to project query and corpus into the same language, which is also reported by Nair et al. (2022). It is possible to leverage the transformers pre-trained with translation language modeling (Chi et al., 2021; Feng et al., 2022) and the established parallel training data (Bonifacio et al., 2021) to improve mAggretriever's cross-lingual retrieval capability, which we leave for future work.

## 6 Conclusion

In this paper, we introduce mAggretriever, an extension of Aggretriever for multilingual retrieval, by combining lexical and semantic features in pre-trained language models for dense retrieval. We propose target-language and self MLM predictions to enhance the efficiency of mAggretriever. Our study highlights the efficiency advantage of self MLM prediction in multilingual retrieval, while target-language MLM prediction offers flexibility for cross-lingual retrieval. Importantly, mAggretriever, solely fine-tuned on English data, demonstrates competitive multilingual retrieval capability compared to other state-of-the-art dense retrievers.

## Limitations

Our research primarily focuses on enhancing multilingual retrieval, specifically targeting monolingual retrieval zero-shot transfer to non-English languages. We plan to extend our study to improve cross-lingual retrieval by leveraging transformers pre-trained with translation language modeling (Chi et al., 2021; Feng et al., 2022). In addition, we only discuss how to improve zero-shot language transferability of dense retrieval. It is possible to further improve model effectiveness by leveraging existing multilingual training data (Bonifacio et al., 2021; Zhang et al., 2021) and better negative mining strategies (Shen et al., 2022). Finally, due to space limitation, we compare mAggretriever with previous state-of-the-art multilingual retrievers on Mr. TyDi (Zhang et al., 2021) in Appendix A.2.

## Acknowledgements

This research was supported in part by the Canada First Research Excellence Fund and the Natural Sciences and Engineering Research Council (NSERC) of Canada. We thank the anonymous referees who provided useful feedback to improve this work.

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

# A Appendix

## A.1 Full Results on MIRACL

Table 5 reports the detailed nDCG@10 and R@100 numbers on MIRACL 15 languages for all the compared models. Note that we do not use English dataset when evaluating on MIRACL.

## A.2 Comparisons on Mr. TyDi

Some previous state-of-the-art multilingual retrievers are evaluated on the test queries of Mr. TyDi (Zhang et al., 2021), the multilingual retrieval dataset similar to MIRACL but with sparse relevance judgements and less languages. In order to compare with the previous models, following the model inference settings in MIRACL, we evaluate our mAggretriever with the backbone of XLM-R. The full results are tabulated in Table 6. We still observe that all the variants of mAggretriever outperform previous state of the art in 6 out of 10 languages. Note that most of the previous retrievers undergo multilingual contrastive learning. For example, mContriever are pre-trained on the corpus with 29 languages while mColBERT are fine-tuned on multilingual MS MARCO dataset introduced by Bonifacio et al. (2021).

Table 5: MIRACL multilingual retrieval comparisons. Bold denotes the best effectiveness.

| Models | ar | bn | es | fa | fi | fr | hi | id | ja | ko | ru | sw | te | th | zh | avg. |
|---|---|---|---|---|---|---|---|---|---|---|---|---|---|---|---|---|
| | | | | | | | **MIRACL Dev** | | | | | | | | | |
| | | | | | | | nDCG@10 | | | | | | | | | |
| (a) BM25 | 48.1 | 50.8 | 31.9 | 33.3 | 55.1 | 18.3 | 45.3 | 44.9 | 36.9 | 41.9 | 33.4 | 38.3 | 49.4 | 48.4 | 18.0 | 39.6 |
| (b) mDPR | 49.9 | 44.3 | **47.8** | **48.0** | 47.2 | 43.5 | 38.3 | 27.2 | 43.9 | 41.9 | 40.7 | 29.9 | 35.6 | 35.8 | **51.2** | 41.7 |
| (c) mContriever | 52.5 | 50.0 | 41.8 | 21.5 | 60.2 | 31.4 | 28.6 | 39.2 | 42.4 | 48.3 | 39.1 | 56.0 | 52.8 | 51.7 | 41.0 | 43.8 |
| (d) Cohere (API) | **61.7** | 59.4 | 23.3 | 47.1 | 63.4 | **46.2** | **49.3** | 44.6 | 46.0 | 49.6 | **46.9** | **61.1** | 61.3 | 54.6 | 36.5 | 50.1 |
| (1) mBERT$_{CLS}$ | 50.9 | 45.2 | 31.7 | 30.5 | 48.5 | 30.5 | 37.7 | 19.8 | 43.4 | 40.0 | 27.0 | 21.2 | 36.4 | 37.7 | 43.9 | 36.1 |
| (2) mBERT$_{AGG}$ | 56.9 | 48.5 | 44.0 | 43.1 | 63.4 | 40.8 | 34.1 | 40.3 | 49.3 | 43.9 | 44.9 | 44.6 | 44.7 | 27.2 | 40.2 | 44.4 |
| (3) mBERT$_{AGG}^{tg}$ | 55.7 | 45.5 | 45.4 | 42.4 | 62.3 | 42.9 | 34.3 | 38.7 | 47.7 | 43.4 | 45.2 | 45.5 | 43.9 | 32.5 | 41.5 | 44.4 |
| (4) mBERT$_{AGG}^{self}$ | 59.4 | 51.0 | 44.6 | 44.5 | 65.3 | 43.6 | 37.4 | 42.1 | 50.2 | 47.8 | 46.3 | 48.5 | 48.0 | 31.1 | 44.3 | 46.9 |
| (5) XLM-R$_{CLS}$ | 46.6 | 46.6 | 29.9 | 43.5 | 44.4 | 28.5 | 41.7 | 31.7 | 40.7 | 45.6 | 27.5 | 22.5 | 50.5 | 53.9 | 35.9 | 39.3 |
| (6) XLM-R$_{AGG}$ | 60.6 | 60.4 | 42.5 | 46.2 | 66.2 | 43.0 | 44.8 | 47.8 | 53.3 | 58.2 | 44.6 | 46.6 | **72.1** | 66.0 | 41.4 | 52.9 |
| (7) XLM-R$_{AGG}^{tg}$ | 60.2 | 60.9 | 44.3 | 46.7 | 65.0 | 43.8 | 48.0 | 47.2 | 53.3 | **59.1** | 45.8 | 45.2 | 70.2 | 67.1 | 42.9 | **53.3** |
| (8) XLM-R$_{AGG}^{self}$ | 61.4 | **61.4** | 42.9 | 46.5 | **66.2** | 41.3 | 46.2 | **48.4** | 53.9 | 57.9 | 46.5 | 47.5 | 71.2 | **66.8** | 41.0 | 53.3 |
| | | | | | | | R@100 | | | | | | | | | |
| (a) BM25 | 88.9 | 90.9 | 70.2 | 73.1 | 89.1 | 65.3 | **86.8** | **90.4** | 80.5 | 78.3 | 66.1 | 70.1 | 83.1 | 88.7 | 56.0 | 78.5 |
| (b) mDPR | 84.1 | 81.9 | **86.4** | **89.8** | 78.8 | **91.5** | 77.6 | 57.3 | 82.5 | 73.7 | 79.7 | 61.6 | 76.2 | 67.8 | **94.4** | 78.9 |
| (c) mContriever | **92.5** | 92.1 | 84.1 | 65.4 | **95.3** | 82.4 | 64.6 | 80.2 | 87.8 | 87.5 | **85.0** | 91.1 | **96.1** | 93.6 | 90.3 | 85.9 |
| (d) Cohere (API) | - | - | - | - | - | - | - | - | - | - | - | - | - | - | - | - |
| (1) mBERT$_{CLS}$ | 84.3 | 81.9 | 67.1 | 84.1 | 61.2 | 72.9 | 76.3 | 47.7 | 80.7 | 71.9 | 61.1 | 50.4 | 75.5 | 69.7 | 90.0 | 71.6 |
| (2) mBERT$_{AGG}$ | 86.9 | 83.8 | 81.0 | 81.0 | 90.2 | 83.2 | 70.9 | 76.6 | 84.1 | 71.7 | 80.8 | 79.4 | 82.4 | 60.0 | 81.3 | 79.5 |
| (3) mBERT$_{AGG}^{tg}$ | 86.2 | 82.2 | 82.1 | 79.9 | 89.7 | 82.9 | 72.6 | 75.7 | 84.3 | 76.3 | 80.5 | 80.9 | 81.7 | 63.8 | 82.6 | 80.1 |
| (4) mBERT$_{AGG}^{self}$ | 88.7 | 85.1 | 80.1 | 82.4 | 92.1 | 86.5 | 77.6 | 80.2 | 85.5 | 78.4 | 82.0 | 81.5 | 83.4 | 65.9 | 84.5 | 82.3 |
| (5) XLM-R$_{CLS}$ | 79.2 | 82.6 | 63.6 | 79.9 | 75.2 | 66.8 | 76.9 | 62.7 | 77.2 | 77.6 | 62.5 | 48.8 | 87.0 | 89.2 | 78.9 | 73.9 |
| (6) XLM-R$_{AGG}$ | 89.8 | 92.5 | 78.8 | 85.4 | 92.6 | 81.8 | 85.1 | 83.5 | 89.0 | 88.5 | 78.9 | 79.1 | 95.4 | 95.1 | 80.6 | **86.4** |
| (7) XLM-R$_{AGG}^{tg}$ | 89.3 | 93.1 | 80.4 | 84.6 | 91.8 | 81.4 | 84.5 | 82.6 | 88.9 | 86.6 | 79.5 | 77.7 | 94.8 | **95.1** | 79.9 | 86.0 |
| (8) XLM-R$_{AGG}^{self}$ | 90.2 | **92.8** | 78.2 | 84.5 | 93.1 | 81.2 | 82.5 | 84.6 | **89.8** | **88.3** | 80.8 | 79.4 | 95.5 | 94.5 | 79.0 | 86.3 |

Table 6: Mr. TyDi multilingual retrieval comparisons with state-of-the-art multilingual retrievers. Bold denotes the best effectiveness. multi CL denotes multilingual contrastive learning.

| Models | multi CL | ar | bn | fi | id | ja | ko | ru | sw | te | th | avg. |
|---|---|---|---|---|---|---|---|---|---|---|---|---|
| | | | | | | **Mr. TyDi Test** | | | | | | |
| | | | | | | MRR@100 | | | | | | |
| (a) BM25 (Zhang et al., 2021) | ✗ | 36.7 | 41.3 | 28.8 | 38.2 | 21.7 | 28.1 | 32.9 | 39.6 | 42.4 | 41.7 | 35.1 |
| (b) mColBERT (Bonifacio et al., 2021) | ✓ | **55.3** | 48.8 | 41.3 | **55.5** | 36.6 | 36.7 | **48.2** | 44.8 | 61.6 | - | - |
| (c) mContriever (Izacard et al., 2022) | ✓ | 43.4 | 42.3 | 35.1 | 42.6 | 32.4 | 34.2 | 36.1 | **51.2** | 37.4 | 40.2 | 39.5 |
| (d) CCP (Wu et al., 2022) | ✗ | 42.6 | 45.7 | 37.2 | 46.2 | 37.7 | 34.6 | 36.0 | 39.2 | 47.0 | 48.9 | 41.5 |
| (e) MSM (Zhang et al., 2023a) | ✗ | 51.6 | 53.0 | 39.4 | 50.5 | 32.0 | 36.8 | 37.2 | 43.4 | 62.6 | 53.5 | 44.7 |
| (1) XLM-R$_{CLS}$ | ✗ | 41.9 | 40.8 | 27.8 | 39.9 | 32.5 | 33.0 | 27.7 | 23.7 | 54.2 | 46.1 | 36.8 |
| (2) XLM-R$_{AGG}$ | ✗ | 52.3 | 55.8 | **43.2** | 55.0 | 40.4 | 40.5 | 41.5 | 45.1 | **77.5** | 57.3 | 50.8 |
| (3) XLM-R$_{AGG}^{tg}$ | ✗ | 52.3 | 55.2 | 43.0 | 54.8 | **41.1** | 40.4 | 44.9 | 46.0 | 76.2 | **58.7** | 51.2 |
| (4) XLM-R$_{AGG}^{self}$ | ✗ | 52.0 | **58.5** | 42.6 | 54.8 | 39.2 | **41.6** | 44.3 | 47.3 | 74.7 | 58.3 | **51.3** |