# OpenReview forum: "mAggretriever: A Simple yet Effective Approach to Zero-Shot Multilingual Dense Retrieval"
_EMNLP/2023/Conference — EMNLP 2023 Main_

### Official Review · Reviewer_MRVN · 2023-07-24

**Soundness:** 4

**Excitement:**

3: Ambivalent: It has merits (e.g., it reports state-of-the-art results, the idea is nice), but there are key weaknesses (e.g., it describes incremental work), and it can significantly benefit from another round of revision. However, I won't object to accepting it if my co-reviewers champion it.

**Paper Topic And Main Contributions:**

This paper extend Aggretriever under multilingual retrieval senarios with two approaches proposed to alliviate the computation cost when incorperate with a large multilingual vocabulary space. One is target-language prediction, which mask the source langugae token during MLM projection. Another is self prediction, which mask all the other tokens except for the input ones. Experiments show that mAggretriever surpasses existing state-of-the-art multilingual dense retrieval models.

**Questions For The Authors:**

- The proposed method demonstrates substantial improvements in MRR and nDCG over mContriever, while showing only a slight improvement in recall. It would be valuable for the authors to provide insights into the reasons for this gap in performance.
- It would be beneficial to include a comprehensive comparison of computation cost and inference latency with other baselines such as mContriever and mDPR to better evaluate the overall efficiency-effectiveness trade-off. Specifically, I would be interested in GPU usage and retrieval latency of mDPR/mContriever comapred to mAggretriever.

**Reasons To Accept:**

- The proposed method is simple, improving both efficient and effective for multilingual retrieval.
- The experimental methodology and analysis conducted in this study are extensive and thorough, with plenty insights and discussion provided

**Reasons To Reject:**

- The motivation behind the Aggretriever-like design is clear, but it seems limited to this specific approach (or say approaches that require lexical representation).
- The comparison made in the paper is not entirely fair, as it only considers Aggretriever-based methods. The efficiency workload stems from the design of Aggretriever which requires the high-dimensional lexical representation, and mDPR/mContriever would not face the such challenge. While the author highlight the effiency of the proposed method, it would be more appropriate to compare the efficiency-effectiveness balance with other methods like mDPR/mContriever.
- [minor] Some notations are undefined and lead to confusion

**Reproducibility:**

3: Could reproduce the results with some difficulty. The settings of parameters are underspecified or subjectively determined; the training/evaluation data are not widely available.

**Reviewer Confidence:**

4: Quite sure. I tried to check the important points carefully. It's unlikely, though conceivable, that I missed something that should affect my ratings.

**Typos Grammar Style And Presentation Improvements:**

- A task formulation of multilingual retrieval is expected.
- Some abbreviations need clarification, such as "tg" (target) and "wp" (word piece), to avoid confusion for readers.

---

> ### Author Rebuttal · Authors · 2023-08-26
>
> We thank you for the insightful discussion and help comments. See below responses.
>
> 1. The comparison is not entirely fair. A comprehensive comparison with mDPR and mContriever should be included:
> First of all, we would like to point out that mAggretriever’s final text representation is the same as other dense retrieval models; i.e., a  768-dimensional single-vector. The only difference is that mAggretriever’s vector is the concatenation of  128-dimensional [CLS] and 640-dimensional agg* vectors, all of which are dense vectors. Thus, there is no latency difference in end-to-end vector search using mAggretriever and other dense models. The only additional efficiency bottleneck of mAggretriever is MLM prediction and compressing the high-dimensional lexical vector into a 640-dimensional vector, which may sacrifice the training cost and query encoding latency as shown in Table1. Furthermore, our most efficient design (i.e., self prediction), due to no MLM prediction required, only shows a minor additional cost on query encoding latency on CPU (this additional cost comes from vector compression). Thus, we believe the comparison of mAggretriever with all the other dense models is fair and comprehensive enough. We will point out this explicitly and thanks for the comments.
>
> 2. The proposed approach is limited; i.e., lexical representation is required:
> The proposed approach is based on pre-trained transformers and as far as we know, all the pre-trained transformer language models can output lexical representations. Thus, we believe our approach can widely apply to different pre-trained transformer backbones.
>
>
> 3. Recall comparison to mContriever:
> There are two main differences between mContriever (with the backbone of mBERT) and our trained models, $\text{mBERT}_{\text{CLS}}$ and $\text{mBERT}_{\text{AGG}}$. First, mContriever experiences contrastive pre-training on 29 languages while our trained models do not; Second, mContriever is further fine-tuned on MS MARCO with a batch size of 1024 while our trained models use a batch size of 96. Although we cannot conduct such experiments to verify the effectiveness of the two factors, we hypothesize the high recall of mContriever comes from the multilingual contrastive pre-training. We thank you for the suggestions and agree that it is worth discussing. We will discuss this point in our revised version.

---

### Official Review · Reviewer_rgyC · 2023-07-31

**Typos Grammar Style And Presentation Improvements:** 1. The self prediction part in Line 1…
**Soundness:** 4

**Excitement:**

3: Ambivalent: It has merits (e.g., it reports state-of-the-art results, the idea is nice), but there are key weaknesses (e.g., it describes incremental work), and it can significantly benefit from another round of revision. However, I won't object to accepting it if my co-reviewers champion it.

**Paper Topic And Main Contributions:**

This paper studies the problem of aggregating [CLS] and MLM softmax predictions for multilingual information retrieval.
In the authors' setting, a multilingual pre-trained model is fine-tuned on English data only
but is evaluated on multilingual data.
To solve this problem,
the authors propose several variants for aggregation,
including whole vocabulary, target language, and self prediction.
Experiments on the MS-MARCO and MIRACL datasets find that the target language aggregation
achieves competitive results while reducing the training and inference costs.


**Questions For The Authors:**

A. How do you determine if a token belongs to the target language? As some tokens may be shared across multiple languages.


**Reasons To Accept:**

1. This paper extends the study scope of Aggretriever from monolingual to multilingual scenarios and proposes several variants for representation aggregation.
2. The experiments are very thorough and some of the results are interesting.


**Reasons To Reject:**

1. Results in Table 2 do not look very high, especially on the MS-MARCO dataset. But this may be due to the fact that the authors do not use mined hard negatives.
2. This paper builds on two assumptions: only English training data is available; aggregation from MLM distribution is necessary. However, these assumptions may not hold in real-world scenarios. Multilingual data are still available in many cases, though they may be much less than English data. Also, many papers utilizing only the [CLS] vector or mean pooling have shown good results.


**Reproducibility:**

4: Could mostly reproduce the results, but there may be some variation because of sample variance or minor variations in their interpretation of the protocol or method.

**Reviewer Confidence:**

4: Quite sure. I tried to check the important points carefully. It's unlikely, though conceivable, that I missed something that should affect my ratings.

---

> ### Author Rebuttal · Authors · 2023-08-26
>
> We thank the reviewer’s insightful discussion. See below responses to the concerns and questions.
>
> 1. Results in Table2 do not look very high in MS MARCO:
> While we agree that knowledge distillation and hard negative mining might further improve our models’ retrieval effectiveness on MS MARCO, our main intention in presenting Table 2 was to provide a fair comparison with all baselines, including, for example, mDPR and mContriever, which only use MS MARCO training data constructed by BM25 mined negatives.
>
> 2. Many works using [CLS] or average pooling have shown good results:
> We admit that the short paper format limits our ability to showcase the full advantages of mAggretriever. Due to these limitations, we cannot compare with every state-of-the-art model, since many of those were evaluated on Mr. TyDi rather than MIRACL. But we report the comparisons in Table 6 (see Appendix A.4). Our mAggretriever consistently beats recent state-of-the-art models, which were trained on either multilingual training data or parallel translation data. We will point this out explicitly in our revised version.
>
> 3. The assumption of only English training data is available is not valid:
> We agree, but, as you mentioned, data from other languages is less plentiful than English, or, in extreme cases, non-existent. In addition, this data, when it is available, is often lower quality than English datasets (e.g. MS MARCO and NQ). Thus, using these non-English datasets would require exploration of how to improve the datasets’ quality as well as sampling strategies for the languages during training. On the other hand, mAggretriever provides an elegant solution to bypass these issues: namely, to train on English data and directly transfer the retrieval performance to other languages. We will clarify this further in our revised version.
>
> 4. How do you determine if a token belongs to the target language?
> Due to space limitations, we describe the implementation details in Appendix A.1 (see target-token set construction). We will add the pointer to the appending in the revised version.
>
> 5. The self prediction part in Line 162-169 is not very clear to me:
> The self prediction is a bag-of-words representation with learned term weights. The learned term weight for each token is $w_i$ in Eq.(3). In our approach section, we mainly discuss how to output the MLM prediction probability; i.e., $p_{q_i}$ in Eq.(3). Since $p_{q_i}$ is normalized by softmax as shown in Eq.(2), directly conducting max pooling over all $p_{q_i}$ in Eq.(3) would ignore the importance of each input token $q_i$. Thus, we also use learned term weights to reweight the probability from each token $q_i$. This design is the same as Aggretriever. Therefore, self prediction is a bag-of-words representation where each appearing token $q_i$ has term weight $w_i$. Thank you for pointing out the unclarity and we will further clarify it in our revised version.

---

### Official Review · Reviewer_xtfQ · 2023-08-03

**Soundness:** 3

**Excitement:**

3: Ambivalent: It has merits (e.g., it reports state-of-the-art results, the idea is nice), but there are key weaknesses (e.g., it describes incremental work), and it can significantly benefit from another round of revision. However, I won't object to accepting it if my co-reviewers champion it.

**Paper Topic And Main Contributions:**

This paper presents mAggretriever, a novel method that efficiently harnesses semantic and lexical attributes from pre-trained multilingual transformers like mBERT and XLM-R for dense retrieval. The authors propose two computationally efficient strategies to approximate MLM prediction. Despite being fine-tuned on English data, mAggretriever demonstrates impressive retrieval performance across various languages.

**Reasons To Accept:**

This paper presents mAggretriever, an extended version of Aggretriever, for multilingual retrieval. To improve the efficiency of mAggretriever, they introduced target-language and self MLM predictions. Although the proposed approaches are straightforward and simple, they demonstrate the effectiveness of the propose methods on two different datasets, using two pre-trained models, namely mBERTR and XLM-R.

**Reasons To Reject:**

As depicted in Table 2, the proposed methods outperform mAggretriever using the whole-vocabulary MLM prediction, namely mBERTAGG and XLM-RAGG.
Lines 147-149 state, "we propose two computationally efficient approaches to approximate MLM prediction."
Since the proposed methods are approximations of the original approach, the question arises: Why do these approximate methods exhibit superior retrieval performance (R@1K, R@100) compared to the original MLM prediction?

**Reproducibility:**

3: Could reproduce the results with some difficulty. The settings of parameters are underspecified or subjectively determined; the training/evaluation data are not widely available.

**Reviewer Confidence:**

4: Quite sure. I tried to check the important points carefully. It's unlikely, though conceivable, that I missed something that should affect my ratings.

---

> ### Author Rebuttal · Authors · 2023-08-26
>
> We thank the reviewer’s insightful questions. See below response.
>
> 1. Why do the approximate methods exhibit superior retrieval performance:
> We can divide the full MLM prediction task into two components: self-prediction and lexical expansion, i.e. predicting tokens other than those explicitly indicated. From the experimental results, we hypothesize that the key element for good transferability into different languages is self-prediction, not lexical expansion. In fact, when conducting full MLM prediction, the lexical expansion may cause errors in the agg* vector. Stated differently, the lexical expansion for English does not transfer to other languages as well as self-prediction of the token itself. We will discuss this further in the revised version.

---

### Meta-Review · Area_Chair_nzYr · 2023-09-07

**Recommendation:** 4

**Metareview:**

All reviewers found merits in the submission. Especially, it extends the study scope of Aggretriever from monolingual to multilingual scenarios. Although the performance improvement is not so great, I think the technical contribution may be inspiring to the EMNLP community.

---

### Decision · Program_Chairs · 2023-10-07

**Decision:**

Accept-Main

**Comment:**

All reviewers found merits in the submission. Especially, it extends the study scope of Aggretriever from monolingual to multilingual scenarios. Although the performance improvement is not so great, I think the technical contribution may be inspiring to the EMNLP community.